# Forecasting influenza in Europe using a metapopulation model incorporating cross-border commuting and air travel

**Sarah C. Kramer** *, **Sen Pei**, **Jeffrey Shaman**

Department of Environmental Health Sciences, Mailman School of Public Health, Columbia University, New York, New York, United States of America

* sck2165@caa.columbia.edu

## Abstract

Past work has shown that models incorporating human travel can improve the quality of influenza forecasts. Here, we develop and validate a metapopulation model of twelve European countries, in which international translocation of virus is driven by observed commuting and air travel flows, and use this model to generate influenza forecasts in conjunction with incidence data from the World Health Organization. We find that, although the metapopulation model fits the data well, it offers no improvement over isolated models in forecast quality. We discuss several potential reasons for these results. In particular, we note the need for data that are more comparable from country to country, and offer suggestions as to how surveillance systems might be improved to achieve this goal.

**Data Availability Statement:** Data on scaled influenza incidence and country-level absolute humidity are available as Supporting Information files, as well as at https://github.com/

## Author summary

In our increasingly connected world, infectious diseases are more capable than ever of rapid spread over large geographical distances. Previous research has shown that human travel can be used to better forecast the transmission of influenza, which may in turn help public health and medical practitioners to prepare for outbreaks with increasing lead time. Here, we developed a model of twelve European countries, in which countries are connected based on rates of commuting and air travel between them. We then used this model, along with publicly available influenza incidence data, to forecast future incidence in Europe. We found that forecasts produced with the network model were not more accurate than those produced for individual countries in isolation. We emphasize the need for aligned influenza data collection practices that are comparable between different countries and which will likely improve forecast accuracy.

## Introduction

In recent years, multiple studies have demonstrated that skillful influenza forecasts can be generated for a range of cities and countries in both temperate [1–8] and tropical [9,10] areas. When operationalized, such forecasts can provide early warning for healthcare workers and

sarahckramer/euro_flu_network. Data on air travel between countries are freely available from Eurostat at: http://appsso.eurostat.ec.europa.eu/nui/show.do?dataset=avia_paocc&lang=en. Commuting data must be requested directly from Eurostat (https://ec.europa.eu/eurostat/help/support).

**Funding:** This research was supported by US National Institutes of Health (NIH; https://www.nih.gov) Grants F31AI138410 (SCK), GM110748 (JS), and T32ES023770 (JS). The funders had no role in study design, data collection and analysis, decision to publish, or preparation of the manuscript.

public health practitioners during both seasonal influenza outbreaks and future influenza pandemics, allowing for a more proactive public health response [11]. Given the high toll of seasonal influenza each year [12,13], as well as the potential for future pandemic emergence [14–16], such proactive responses, informed by real-time forecasts, could reduce influenza-related morbidity and mortality.

Most forecasting to date has been generated for individual locations. City- and country-level outbreaks, however, do not occur in isolation. Research shows that human travel, ranging from long-distance air travel [17–20] to short-distance commuting [21–25], influences how outbreaks of infectious disease spread. Recent work suggests that, by representing human movement between locations in disease models, forecasts of influenza can be significantly improved at the borough level in New York City [26] and at the state level in the United States [27]. In particular, prediction of local outbreak onset, defined as the first of three consecutive weeks influenza incidence rises above a critical threshold, is difficult to accurately forecast for a single location in isolation and can be greatly improved by considering commuting and other travel between states [27].

Europe represents an interesting opportunity to test the utility of similar models for forecasting international, rather than interstate, influenza transmission. The Schengen Agreement allows citizens and residents of participating countries to cross international borders freely and without border checks, including for employment [28]. Thus, although Europe is made up of multiple sovereign countries, commuting and other forms of travel between these countries remain high.

We recently showed that freely available data from the World Health Organization (WHO) can be used to produce skillful forecasts of influenza activity for many European countries in isolation [29]. The logical next step, then, is to see whether considering human movement between these countries can yield improvements in forecast accuracy, as was observed at the state level within the United States.

Here we describe the development of a network model of influenza transmission among twelve European countries (Austria, Belgium, Czechia, France, Germany, Hungary, Italy, Luxembourg, the Netherlands, Poland, Slovakia, and Spain), incorporating both cross-border commuting and air travel. We then generate retrospective forecasts of influenza activity using the network model, and compare forecast accuracy to isolated, country-level forecasts. We hypothesize that, by incorporating observed human movement between countries, we will significantly improve forecast accuracy, particularly for onset timing predictions.

## Materials and methods

### Influenza data

Influenza data were obtained from the World Health Organization's (WHO) FluNet [30] and FluID [31] surveillance platforms, which collect virologic and clinical data, respectively, from WHO member countries. Clinical data are reported as numbers of either influenza-like illness (ILI) or acute respiratory infection (ARI), depending on country and season (see S1 Text). Because these measures are based on symptoms that are not specific to influenza, they include cases caused by infections with other respiratory pathogens. To remedy this lack of specificity, we multiplied weekly cases of ILI/ARI by the proportion of tests for influenza that were positive during the same week, as reported in the FluNet data. We refer to this measure as syndromic+ [29].

Although circulating influenza types and (sub)types are often similar between European countries, substantial differences are sometimes observed (see S1 Text). For this reason, we calculated (sub)type-specific syndromic+ cases for each country by multiplying ILI/ARI cases by

the proportion of tests positive for H1N1, H3N2, or B influenza. Tests positive for unsubtyped influenza type A in a given country were assigned as H1N1 or H3N2 in proportion to the respective rates of each subtype that week.

For this study, we focus on seasonal influenza. Thus, data were downloaded and processed for the 2010–11 through 2017–18 seasons, where a season is defined as beginning in calendar week 40 (i.e. late September/early October) and ending in calendar week 20 (i.e. mid-May) of the following year. Further information on how influenza data were processed can be found in [29]. Data are visualized in S1 Fig, and fully formatted and scaled influenza data for (sub)types H1N1, H3N2, and B can be found as S1, S2, and S3 Datasets, respectively.

## Travel data

Data on both air travel and commuting between European countries were obtained from Eurostat, the European Union's (EU's) statistical office [32]. Specifically, monthly data on the number of passengers carried by aircraft departing from one European country and arriving in another from 2010 through 2017 were obtained from [33], while yearly data on the number of individuals living in one country and working in another for each country pair from 2010 through 2017 were obtained from the Labour Force Survey [34].

Monthly air travel data were averaged over all available years to yield average travel flows by month, and these data were converted to daily rates. In order to hold model populations constant by country, the travel matrix was made symmetric by averaging the travel rates in both directions along each route.

Unlike the air travel data, commuting data were not available for every possible route. Eurostat calculates country-specific thresholds below which data are either not reported (threshold a) due to low reliability and concern for anonymity, or are reported with a note concerning their reliability (threshold b). We included only countries that had at least one incoming and at least one outgoing route with reliable data (i.e. above both thresholds) in the model. Because commuting rates along some routes changed substantially over time, we formed "seasonal" commuting matrices by taking the mean of the two yearly matrices contemporary with each influenza season (e.g., for the 2010–11 season, we averaged commuting rates for 2010 and 2011), rather than using an average over all years. Data from 2017 alone were used when forecasting the 2017–18 influenza season. Finally, routes where data were suppressed or unavailable were filled using a random value between 0 and the relevant country-specific threshold a, obtained using Latin Hypercube sampling over all seasons and routes. Thus, the number of commuters along these routes varied across each of the 300 ensemble members used during forecasting (see below), providing a stochastic distribution of possible rates.

Based on the availability of good-quality influenza, air, and commuting data, we included 12 European countries in our model (Fig 1A). More detailed information on data processing can be found in S1 Text.

## Humidity data

Absolute humidity data were obtained from NASA's Global Land Data Assimilation System (GLDAS) [35]. Values were available every three hours and at a spatial resolution of 1˚x1˚ for 1989–2008. We formed 20-year climatologies for each country by aggregating data to the daily level and averaging each daily value over twenty years. Climatologies were then aggregated to the country level by averaging the climatologies for each individual 1˚x1˚ grid cell within a country, weighted by the number of people living within that grid cell. We weighted by population size in order to better estimate climatic conditions in areas where more people lived, and where more people would therefore be spreading influenza. A more detailed description

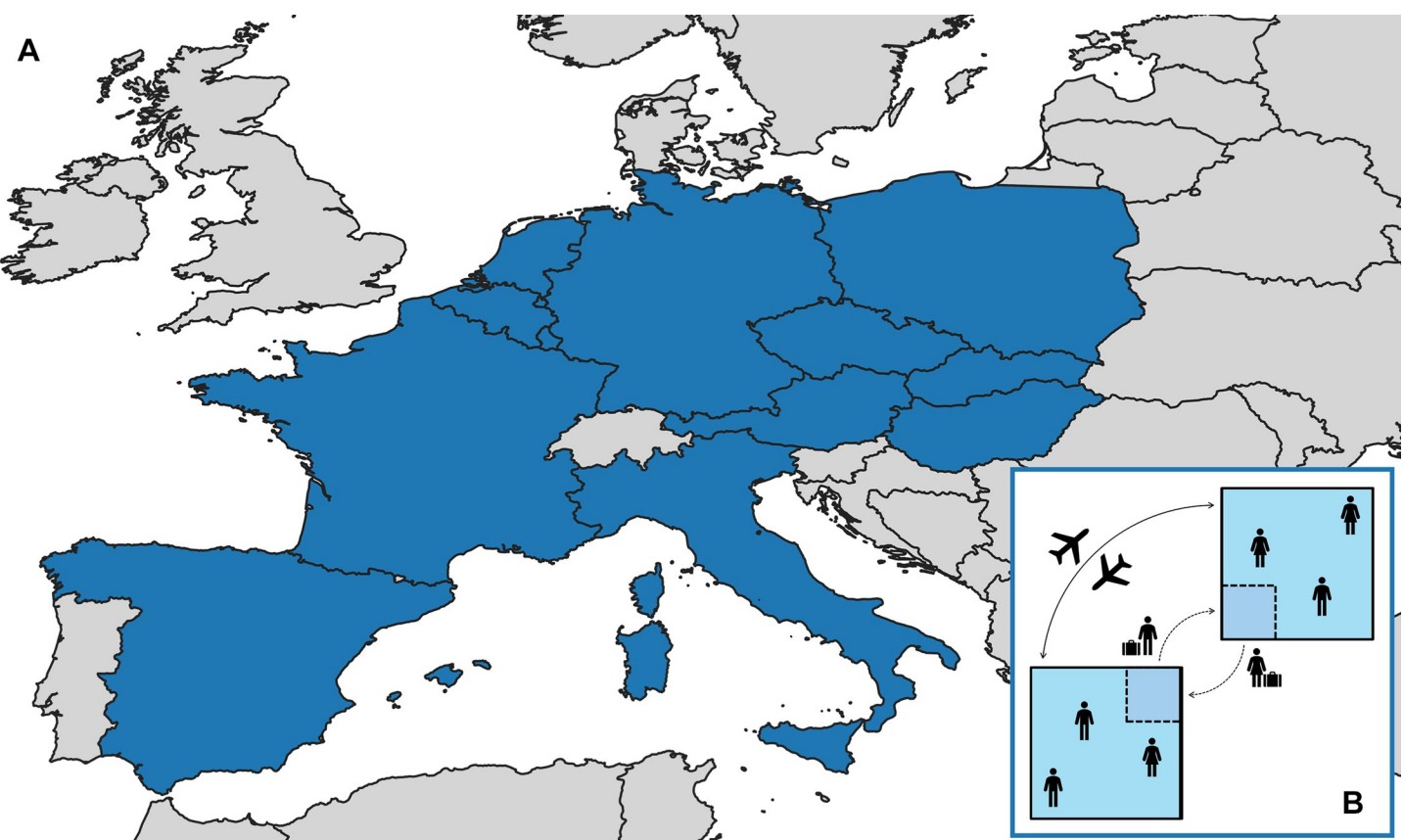

**Fig 1. Network model development.** (A) A map of the twelve countries with good-quality influenza and travel data used in the network model. (B) A schematic of a network model with 2 countries and 4 subpopulations. Large squares represent countries, and smaller squares represent residents who commute internationally. The dotted borders on these smaller squares represent their ability to randomly mix with their work country during the day, and their home country at night. Arrows with dotted lines represent asymmetric daily commuting flows, and the solid, double-headed arrow represents symmetric random air travel.

of these data and how they were processed can be found in [29], and the processed climatologies are available as S4 Dataset.

## Network model

We constructed a networked metapopulation model for the 12 countries, in which subpopulations were assigned to compartments based on both their country of residence and the country in which they work. Within each country, influenza transmission is modeled according to a simple, humidity-forced SIRS model assuming homogenous mixing:

$$\frac{dS}{dt} = \frac{N - S - I}{L} - \frac{\beta(t)IS}{N}$$

$$\frac{dI}{dt} = \frac{\beta(t)IS}{N} - \frac{I}{D} \tag{1}$$

In the above equations, $N$ is the country's population; $S$ and $I$ are the number of people susceptible to and infected with influenza, respectively; $t$ is the time in days; $\beta(t)$ is the rate of transmission at time $t$; $L$ is the average time before immunity is lost; and $D$ is the mean infectious period. The number of newly infected people, $newI$, at time $t$ is calculated as $\beta(t)I_tS_t$ over

$N$. The basic reproductive number, $R_0$, at time $t$ is equal to $\beta(t)$ times $D$, and humidity-forcing of transmissibility is modeled by calculating $R_0$ as:

$$R_0(t) = e^{-180q(t) + \ln(R_{0_{diff}})} + (R_{0_{max}} - R_{0_{diff}}) \qquad (2)$$

where $R_{0max}$ is the maximum possible daily basic reproductive number, $R_{0diff}$ is the difference between the maximum and minimum possible basic reproductive number, and $q(t)$ is the absolute humidity on day $t$ [2]. Past work has demonstrated that virus survival and transmissibility between guinea pigs decreases exponentially with increasing absolute humidity [36]. However, the exact extent to which this impacts person-to-person transmission remains unknown. $R_{0diff}$ in particular allows modulation of the effect of humidity during a given outbreak. Because the model assumes a single pathogen, we assume that the model parameters $L$, $D$, $R_{0max}$, and $R_{0diff}$ are the same for all countries.

In addition to within-country transmission, individuals are permitted to travel between countries by two methods. First, individuals are assigned to be commuters between given country pairs according to Eurostat data. These individuals spend daytime (eight hours, or one-third of each day) in their designated work country, and nighttime (16 hours) in their country of residence. Commuters mix homogenously with the entire population of the country in which they currently are. Commuting is assumed to be a daily occurrence; thus, commuting in the model captures repeated travel done by the same individuals over time. Because commuting populations are explicitly modeled, we permit the commuting network to remain asymmetric.

Any individual in the model population may also travel to another country randomly, according to observed monthly rates of air travel between European countries. Unlike commuters, "random" travelers are not tracked, and return travel is not explicitly modeled. Furthermore, while commuting is a daily occurrence, random travel is a one-time event. To maintain stable population sizes, daily air travel rates between each country pair are averaged, and this average is used as the rate of travel in both directions. Random travelers may introduce new infections to a country through their movement, but any propagation of the outbreak occurs solely through the SIRS dynamics. An additional parameter, $air_{Adj}$, is multiplied by the entire air travel matrix prior to model integration, and allows for the possibility of a small amount of error in the air travel data.

Because we are primarily interested in the ability of information on international travel to improve forecast accuracy, no random seeding of new infections throughout the outbreak is incorporated. The full equations describing the modeling process can be found in S1 Text, and a simplified schematic of the model can be seen in Fig 1B.

## Retrospective forecast generation

We performed retrospective forecasts of (sub)type-specific influenza activity using the network model and influenza data described above in conjunction with a Bayesian data assimilation method. This process consists of a fitting and a forecasting step.

*Model Fitting*: First, we initiate an ensemble of 300 model runs by drawing from realistic parameter ranges ($0.4N < S < 0.9N$, $0 < I < 0.00005N$, $2.0 < R_{0max} < 2.8$, $0.2 < R_{0diff} < 1.0$, $2.0$ days $< D < 7.0$ days, $1095$ days $< L < 3650$ days, $0.75 < air_{Adj} < 1.25$) using Latin Hypercube sampling, which allows for the efficient exploration of the state and parameter space. We then integrate the network model, described above, forward in time. At each week, we halt the integration and adjust the observed model state variables (i.e. aggregated *newI* for each country), as well as unobserved model state variables (i.e. $S$, $I$, and *newI*) in all 144 compartments along with the system parameters ($R_{0max}$, $R_{0diff}$, $D$, $L$, and $air_{Adj}$) using the Ensemble

Adjustment Kalman Filter (EAKF), a Bayesian data assimilation method commonly used in weather and influenza forecasting [2,27,37]. This adjustment is performed by assimilating observations from each of the twelve countries sequentially through application of Bayes Rule:

$$p(X_t|O_{1:t}) \propto p(X_t|O_{1:(t-1)}) \cdot p(O_t|X_t) \tag{3}$$

where $p(X_t|O_{1:(t-1)})$ is the prior distribution of the modeled cases per 100,000 population in a country given all observations up to but not including time $t$, $p(O_t|X_t)$ is the likelihood of the observed incidence per 100,000 population at time $t$ given the modeled incidence rate at time $t$, and $p(X_t|O_{1:t})$ is the posterior distribution of the modeled rate at time $t$ given all observations thus far. More specifically, $X_t$ consists of the observed state variables, in other words the modeled country-level incidence per 100,000 population in each of the 12 countries included in the model, at time $t$; while $O_t$ consists of the scaled observations (the syndromic+ data described under "Influenza Data" after applying the scaling factors described below) for each of the 12 countries. Since the model keeps track of the number of newly infected individuals per compartment, values in $X_t$ are calculated by aggregating model incidence over all compartments containing populations that live in a given country. In other words, for a given country, $i$, at time $t$:

$$X_{t,i} = \sum_n newI^i_{t,n} \tag{4}$$

where $newI^i_{t,n}$ is the modeled influenza incidence at time $t$ among those who live in country $i$ and work in country $n$. Then, the number of susceptible ($S_{t,i}$), infected ($I_{t,i}$), and newly infected ($newI_{t,i}$) people in each individual compartment, as well as the model parameters ($R_{0max}$, $R_{0diff}$, $D$, $L$, and $air_{Adj}$), are updated according to cross-ensemble covariability with the observed state variables (i.e., the country-level rates). In this way, model states and parameters are trained based on the observations of influenza incidence using the EAKF, with the networked SIRS model acting as the forward operator. We note that, while the original bounds on model states and parameters are chosen to reflect biologically plausible values, the EAKF is capable of exploring values outside of these ranges, and states and parameters are therefore not strictly constrained by these initial bounds.

As the outbreak progresses and successive observations are fit, there is a tendency for the variance between the ensemble members to shrink, potentially leading to filter divergence, in which the ensemble error variance becomes so low that observations are essentially no longer considered in the fitting process [37]. We attempted to prevent divergence by multiplicatively inflating the prior model variance by 1.05 at each time step, prior to assimilation of observations [1,27]. When assessed empirically, we found little evidence of filter divergence during the time periods examined in this work for most outbreaks (S2 Fig).

We use ensembles of size 300 based on prior work suggesting that the use of additional ensemble members does not improve the performance of the SIRS-EAKF system. To ensure that the system is capable of adjusting model states and parameters accordingly, we conducted sensitivity analyses exploring the correlations between values in $X_t$ and various unobserved model states and parameters, as well as assessing state and parameter fits in synthetic tests (see S1 Text for details, as well as S3, S11 and S12 Figs, and S4 Table). Further details on the EAKF and its use in influenza forecasting can be found in S1 Text and in [2,27,37,38].

*Forecasting*: After model fitting, forecasts are produced by taking the most recently inferred model states and parameters and running them forward in free simulation until the end of the season. We generate weekly forecasts for each season for calendar weeks 44 through 69 (i.e., late October through mid April), such that forecasts are generated throughout the outbreak.

To assess the effects of stochasticity during model initiation, five separate runs, each consisting of 300 ensemble members, are performed.

## Choice of seasons

While extensive cocirculation of influenza (sub)types is common, outbreaks tend to be dominated by one or two (sub)types. To avoid forecasting a given (sub)type during a season where it has little impact, we used a criterion that could also be applied to forecasting in real time. Specifically, forecasting only took place if the positivity rate for a (sub)type of interest exceeded 10% for three consecutive weeks in at least four of the twelve countries. We therefore forecasted influenza H1N1 for seasons: 2010–11, 2012–13, 2013–14, 2014–15, 2015–16, 2017–18; H3N2 for seasons: 2011–12, 2012–13, 2013–14, 2014–15, 2016–17; and B for seasons: 2010–11, 2012–13, 2014–15, 2015–16, 2016–17, 2017–18.

## Choice of scaling factors

Model fitting using the EAKF, as described above, relies on both $X_t$ and $O_t$ representing influenza incidence rates per 100,000 population. Our raw data, however, are count data. As most countries do not report reliable or consistent data on the number of patient visits, syndromic+ rates per 100,000 population cannot be directly calculated. However, in past work we have observed that our model systems perform optimally when attack rates fall between 15–50% of a model population size within a season. We therefore calculated scaling factors, $\gamma$, for each country and (sub)type by first calculating the range of values that yield attack rates between 15% and 50% for each season, $i$, ($[\gamma_{15,i},\gamma_{50,i}]$). In order to avoid overfitting, scaling factors for each individual season were calculated based on the attack rates for all other seasons of a given country and (sub)type. Thus, season-specific scaling factors for each country and (sub)type, for a given season $j$, were selected based on the rule:

$$\gamma_j = \begin{cases} if \ \exists \gamma \in \mathbb{R}: \ \gamma_{15,i} < \gamma < \gamma_{50,i} \forall i \neq j: & max_{i=1}^n(\gamma_{15,i\neq j}) \\ else: & min_{i=1}^n(\gamma_{50,i\neq j}) \end{cases} \quad (5)$$

Values in $O_t$, then, are calculated by multiplying the syndromic+ data for each country, (sub)type, and season by the relevant scaling factor, and we treat these scaled observations as the estimated syndromic+ rate per 100,000. For each (sub)type, only the seasons listed above in the section "Choice of Seasons" were considered when calculating scaling factors. Additionally, to avoid implausibly high attack rates per 100,000 population, if the highest scaling factor for a given country and (sub)type exceeds the next-highest value by more than 150%, we reduce it to equal 1.5 times the next-highest value. We previously used a similar method to calculate scaling factors in our work generating individual country-level forecasts for multiple countries throughout the world, and found forecasting with the resulting scaled observations feasible [29]. Scaling values by country, (sub)type, and season are presented in S1 Table.

## Individual country forecasts

Forecasts for individual countries in isolation were generated as described in [29]. As with the network model, isolated country-level models were trained through assimilation of 5 to 30 weeks of observations using the EAKF, and forecasts were generated using the most recently updated model states and parameters. Because fitting up through the forecast start week was performed for each country in isolation, forecasts could not be produced at weeks where no observation was available for a given country. Isolated country forecasts used scaling factors as calculated in the previous section. Because no travel between countries is included in the

individual models, we seeded new infections in each country at a rate of 0.1 cases per day, or one new case every ten days. Furthermore, because each country was fit independently, model parameters were allowed to differ by country, unlike in the network model.

## Historical forecasts

Although our primary objective in this work is to compare forecasts produced by the metapopulation and isolated models, it is also important to assess whether our mechanistic models outperform simple forecasts based on historical expectance. For each country and (sub)type, baseline probabilistic forecasts of peak timing, peak intensity, and onset timing for a given season were taken to be the distribution of observed peak timing, peak intensity, and onset timing values among all other seasons. Log scores for each country, (sub)type, and season were then calculated as the natural logarithm of the probability assigned to the observed value. For peak intensity, values were binned as described in the following section.

## Forecast assessment

We assessed system ability to forecast the timing and magnitude of outbreak peaks (peak timing and intensity, respectively), as well as the timing of outbreak onsets. We define outbreak onset as the first of the first three consecutive weeks to exceed 500 scaled cases, as in our past work using these data [29]. For each forecast, we calculated the log score for each of these three metrics by assigning individual ensemble members to bins of size one week (for peak and onset timing) or of size 500 scaled cases (for peak intensity), then calculating the natural logarithm of the proportion of ensemble members falling into the same bin as the observed value. Forecasts for which none of the 300 ensemble members fell within the same bin as the observed value were given a score of -10. The log score has the advantage of being a strictly proper scoring rule, meaning that the expected value has a unique maximum when the forecast distribution matches the true distribution [39,40]. Additionally, it has been commonly used in past forecasting work, including in the United States Centers for Disease Control and Prevention's (CDC) Predict the Influenza Season Challenge [41].

Because the network model considers information from several countries, it is capable of generating forecasts for a given country even at timepoints at which that country has no reported observation, unlike an isolated model. To ensure fairness when comparing between the network and isolated models, we considered only those forecasts which are generated by both models.

## Forecast comparison

Although our analysis is not, strictly speaking, paired, we generated forecasts using both the network and isolated models for the same countries, seasons, (sub)types, and weeks. To take advantage of this design, we performed statistical analyses by observed lead week (i.e., the difference between the week of forecast generation and the observed peak or onset week), rather than predicted lead week, despite the fact that observed lead week cannot be known in real-time. Specifically, we used the Friedman test [42] to compare log scores for peak timing, peak intensity, and onset timing between the network and isolated models for forecasts initiated between six weeks prior to and four weeks after the observed peak/onset. This is a window of time that is of interest to public health practitioners using forecasts to inform decision making.

First, countries with no observed outbreak for a given season and (sub)type are removed from consideration for that specific season-(sub)type combination, as our focus is on forecasts of key outbreak timepoints that are not defined when no outbreak occurs. Of 198 possible country-season-(sub)type combinations, 29 (14.65%) have no observed onset. Second, we

remove forecasts where no onset is predicted from consideration. This is because our outcomes of interest (peak timing, peak intensity, and onset timing) are not defined for a forecast predicting that no outbreak will occur. However, this decision also has practical relevance: we have seen in past work that forecasts predicting no onset also tend to have wide credible intervals, making it difficult to meaningfully use them in public health decision making. Because the ultimate goal of forecasting work is to produce systems capable of informing decision making, we focus on the forecasts (i.e., those with a predicted onset) likely to be considered in such applications.

The Friedman test works by ranking the two model forms for each country-season-(sub) type-week pair, then determining whether either model outperforms the other significantly more often. Therefore, country-season-(sub)type-week pairs where either the network or the isolated model (or both) produce no forecasts predicting an onset were removed. We report the number of pairs removed by observed lead week in S2 Table. We emphasize that, when a forecast is removed, the entire 300-member ensemble is removed from consideration; individual ensemble members are never removed. Then, because the five model runs for each of these pairs (see Retrospective Forecast Generation above) are not independent, we randomly chose a single run for each country-season-(sub)type-week before conducting the Friedman test. This process was repeated 1000 times, and the median p-value was used.

## Code availability

The code used to generate and evaluate model fits and forecasts can be found at: https://github.com/sarahckramer/euro_flu_network.

## Results

### Influenza data

Both influenza and travel data were available for a total of 12 countries (Fig 1A). Nine countries had good-quality [29] data for all 8 seasons, two for 7, and one for 6. Outbreaks occurred in between 2 and 12 countries (mean = 9.94, median = 11.0) during each (sub)type-specific seasonal outbreak. Within a given outbreak, the time between the latest and earliest onset ranged from 2 to 12 weeks (mean = 7.65, median = 7.0); differences in peak timing ranged from 2 to 13 weeks (mean = 7.88, median = 8.0). Descriptive statistics broken down by (sub)type can be found in S1 Text.

### Travel networks

*Air travel*: In general, rates of air travel were higher in western than eastern Europe. On average, rates of daily air travel were highest in and out of Germany and Spain, while the lowest rates were observed in and out of Slovakia and Luxembourg. The mean daily number of passengers was 2813.57, compared to a median of 741.98, indicating that travel rates were mostly low, with a small number of routes (i.e., country pairs) having particularly high rates. In general, rates of air travel are higher in summer, and lower in winter.

*Commuting*: For a given season, commuting data between countries were available for between 46 and 58 of 132 possible routes. On average, the number of commuters was by far the highest from France to Luxembourg, and from Poland to Germany. Net commuter inflow was highest for Luxembourg, Germany, and Austria; and net outflow was highest for France, Slovakia, and Poland. Commuting rates generally increased or remained steady over time, with the greatest growth observed among commuters from Slovakia, Czechia, and Spain into Germany. For routes where data in both directions were reported, average commuting rates

were highly asymmetric, with travel in one direction on average 12.64 (median = 2.78) times higher than in the other. (As explained above in the Materials and Methods, asymmetric commuting flows were permitted in the model.)

## Model fit

In general, the network model was capable of closely fitting the observations from all countries (overall Pearson's correlation coefficients had median = 0.958 and mean = 0.913; A(H1) only: median = 0.968, mean = 0.944), despite substantial differences in the intensity of (sub)type activity (Fig 2). Fits for all five model runs are very similar, suggesting that the model-inference system is not particularly sensitive to initial conditions. Similar plots for the same season for (sub)types A(H3) and B (S4 Fig), demonstrate that fit quality remained high (Pearson's correlation coefficients for A(H3): median = 0.955, mean = 0.923; for B: median = 0.945, mean = 0.876), but that the model sometimes has trouble fitting late-season increases in intensity (see S4B, S4D and S4E Fig), likely due to filter divergence over time (see S2A Fig). Over all seasons, model fit by RMSE and correlation coefficients varied significantly both by (sub)type and country (Kruskal-Wallis tests, p < 0.0002). Post-hoc Nemenyi tests revealed that fit quality was significantly highest for A(H1) by both RMSE and correlation coefficients (p < 0.01 for all

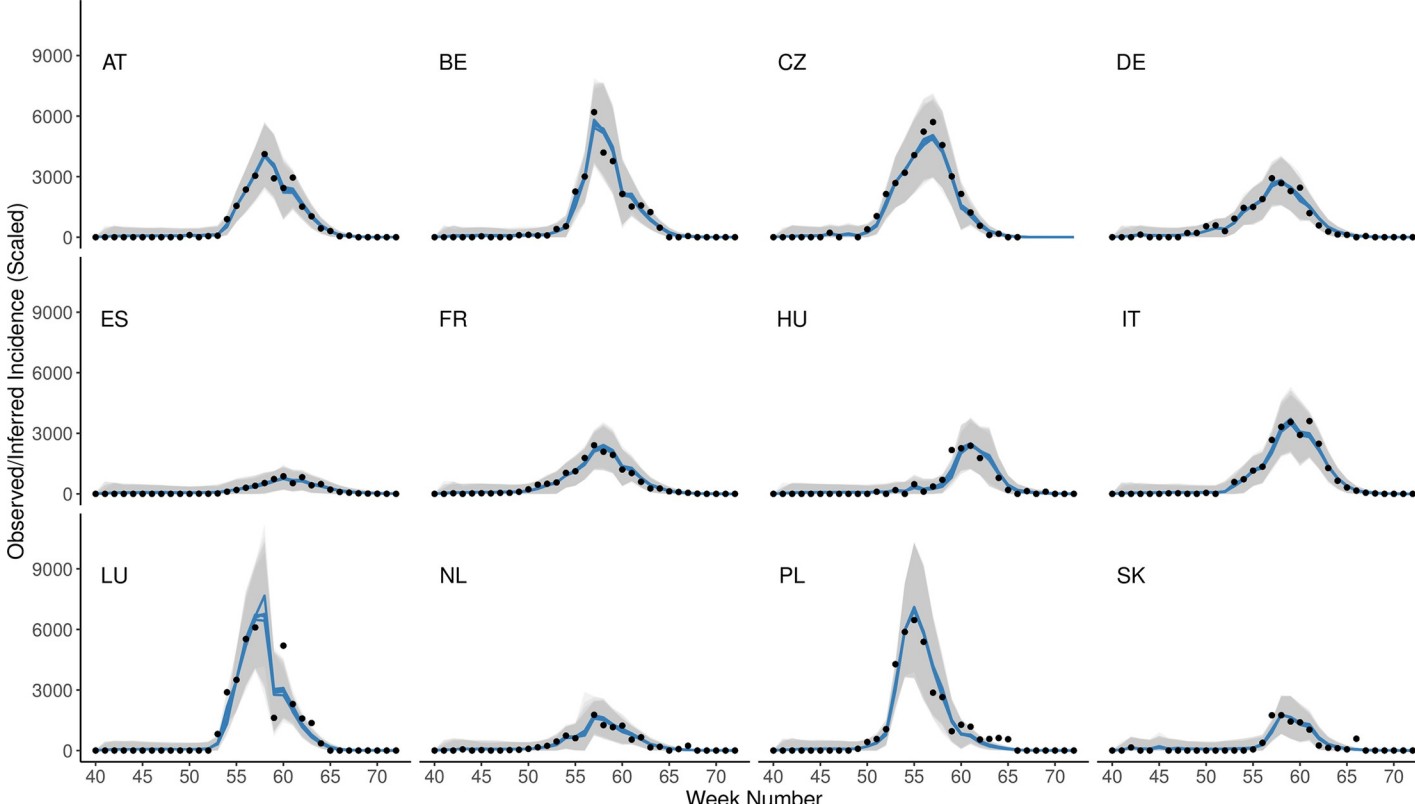

**Fig 2. Model fit to influenza observations.** Scaled observed syndromic+ observations ($O_{1:33}$) for A(H1) throughout the 2012–13 season are shown as points; blue lines represent inferred model incidence ($X_{1:33}$). Note that, because each timepoint is fit based on activity in all countries with observations, fits are shown for a country even for weeks where that particular country had no available observation. Each line represents the mean inferred model incidence over 300 ensemble members for one of five model runs, each with different starting conditions; 95% confidence intervals (calculated using the ensemble mean, $\bar{X}$, and standard deviation, $\sigma$, as $\bar{X} \pm 1.96 \cdot \sigma$) for each run are shown in gray. The root mean square error (RMSE) for this season and (sub)type varied between 60.36 (ES) and 642.99 (LU), and the mean RMSE over all countries was 241.33 (median = 200.09). Pearson's correlation coefficients ranged from 0.920 (HU) to 0.993 (CZ), with a mean correlation coefficient of 0.973 (median = 0.982).

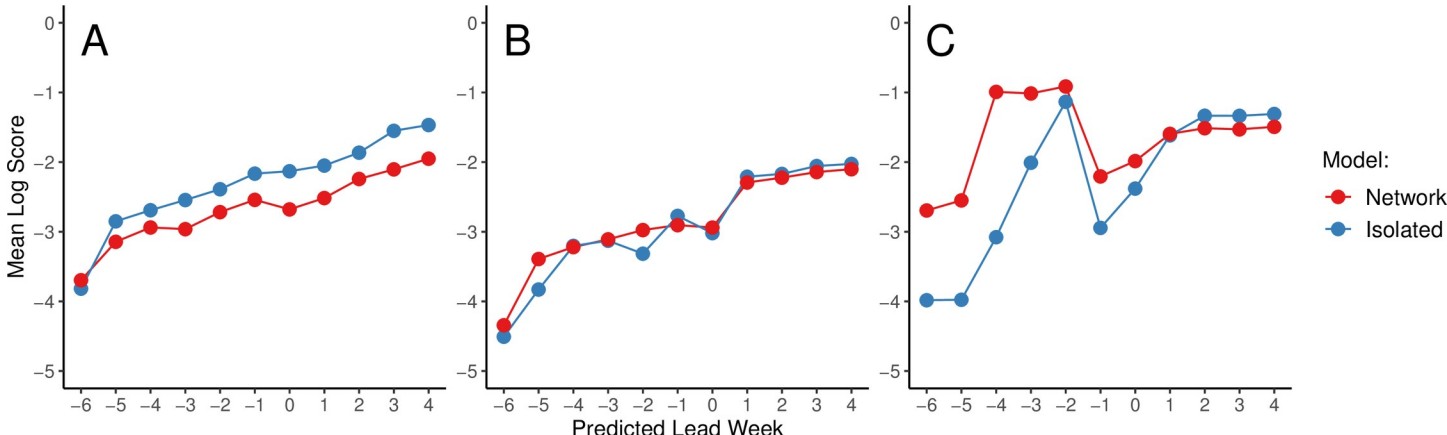

**Fig 3. Retrospective forecast accuracy by predicted lead week.** Mean log scores for forecasts generated using the network (red) and isolated (blue) models are shown by predicted lead week for peak timing (A), peak intensity (B), and onset timing (C).

comparisons), and that fit quality was lowest for A(H3) when measured by RMSE ($p < 0.03$). When assessed by country, only Luxembourg significantly underperformed in comparison to most other countries (using a Bonferroni-corrected p-value cutoff of $0.01/66 = 0.00015$ to account for multiple comparisons); this was true when considering both RMSE and correlation coefficients. Information on inferred state and parameter values can be found in S1 Text and S14 Fig.

## Retrospective forecast accuracy

The mean log scores for predictions of peak timing, peak intensity, and onset timing by predicted lead week and by (sub)type are shown in Fig 3A–3C. Because the timepoint relative to the peak cannot be known in real time, we plot forecast accuracy by predicted lead week, i.e. the difference between the week at which a forecast was initiated and the predicted peak week. As discussed under "Forecast Comparison" above, any forecast not predicting an onset (i.e., where more ensemble members predicted no onset than any single week value) was considered to be unreliable (i.e., unlikely to produce accurate and well-calibrated estimates of peak timing and intensity) and excluded. (However, we briefly explore the model's ability to forecast whether or not an onset will occur in S1 Text and S13 Fig) The number of forecasts produced at each predicted lead week that also predicted any outbreak onset are shown in Table 1.

We found that the isolated model outperformed the network model for peak timing, especially as the predicted peak approached and was passed (Fig 3A), and that the two models performed similarly for peak intensity (Fig 3B). However, we observed a slight improvement in the mean log score for onset timing prior to predicted onset, and especially at predicted leads of -6 to -3 weeks (Fig 3C). Additionally, we note that the network model produced

**Table 1. Number of forecasts predicting any onset by predicted peak and onset lead weeks.**

|  |  | -6 | -5 | -4 | -3 | -2 | -1 | 0 | 1 | 2 | 3 | 4 |
|---|---|---|---|---|---|---|---|---|---|---|---|---|
| Peak Lead Week | *Network* | 339 | 548 | 699 | 744 | 569 | 435 | 581 | 789 | 812 | 798 | 787 |
|  | *Isolated* | 157 | 534 | 701 | 646 | 552 | 539 | 650 | 879 | 842 | 799 | 781 |
| Onset Lead Week | *Network* | 2 | 16 | 11 | 32 | 57 | 129 | 705 | 776 | 788 | 777 | 781 |
|  | *Isolated* | 11 | 43 | 20 | 47 | 44 | 48 | 626 | 717 | 755 | 769 | 782 |

substantially more forecasts than the isolated models at predicted lead -1, although this pattern does not hold for earlier lead weeks (Table 1). When assessed by observed lead week, the same general patterns emerge (S5A–S5C Fig).

Similar results were observed when forecasts were paired by season, country, subtype, and observed lead week, and only those pairs for which both the network and isolated models predict any onset were maintained (S5D–S5F Fig). Friedman tests on these results revealed that none of the apparent differences between the network and isolated models observed in Fig 3 and S5 Fig were statistically significant (all p > 0.05).

When results were separated by influenza (sub)type and assessed by predicted lead week, the network model only had more accurate early forecasts of onset timing for A(H1); however, one week before the predicted onset, the network model produced more forecasts of onset for all (sub)types (S6 Fig and S3 Table). As with the overall results, no significant differences between the network and isolated models were identified when Friedman tests comparing (sub)type-specific forecasts were conducted.

Because so few forecasts are generated before the predicted onset, meaningful comparisons of the results for onset timing by country are not possible, and we have consequently not plotted these results. Log scores for peak timing and intensity by country are presented in S7 Fig, and are discussed briefly in S1 Text.

Forecasts produced using both the metapopulation and isolated models consistently and substantially outperformed forecasts based on historical expectance alone. Mean log scores achieved by historical forecasts were -7.95 for forecasts of peak timing, -7.79 for peak intensity, and -7.45 for onset timing. No significant pairwise differences between (sub)types were observed (Kruskal-Wallis and post-hoc Nemenyi tests).

Forecast comparisons based on an alternative accuracy metric, specifically mean absolute (percentage) error, or MA(P)E, are displayed overall and by (sub)type in S8 Fig and S9 Fig, respectively, and are briefly discussed in S1 Text.

## Forecast calibration

For a forecast to be useful, it needs to communicate not only a prediction, but also a level of certainty for that prediction. Although the log score considers both sharpness and calibration, we also assessed calibration in isolation by calculating how often observed metrics fell within various prediction intervals implied by ensemble spread. As in Fig 3, results are shown by predicted lead week; we focus here only on predictions made before the predicted peak or onset week. Forecasts of peak timing and intensity appear to be well-calibrated for both the network and isolated models, and forecasts of peak intensity using the network model appear to be slightly better-calibrated than those generated in isolation (Fig 4A, 4B, 4D and 4E). The isolated models seem to produce better-calibrated forecasts of onset timing, except at very early leads (Fig 4C and 4F). However, it is important to note that the sample size here is very small, as not many forecasts were produced prior to outbreak onset (see Table 1). An alternative measure of forecast calibration is explored in S1 Text and S10 Fig.

## Discussion

Here we describe the development of a network model for influenza transmission in Europe, and test whether the network model, by incorporating human travel between countries, improves upon isolated country models in forecasting influenza activity. We found that the network model tended to produce more accurate forecasts of onset timing at early predicted leads, and generated substantially more forecasts at predicted leads of -1, indicating that the model can better anticipate when an onset will occur in the following week. However, the

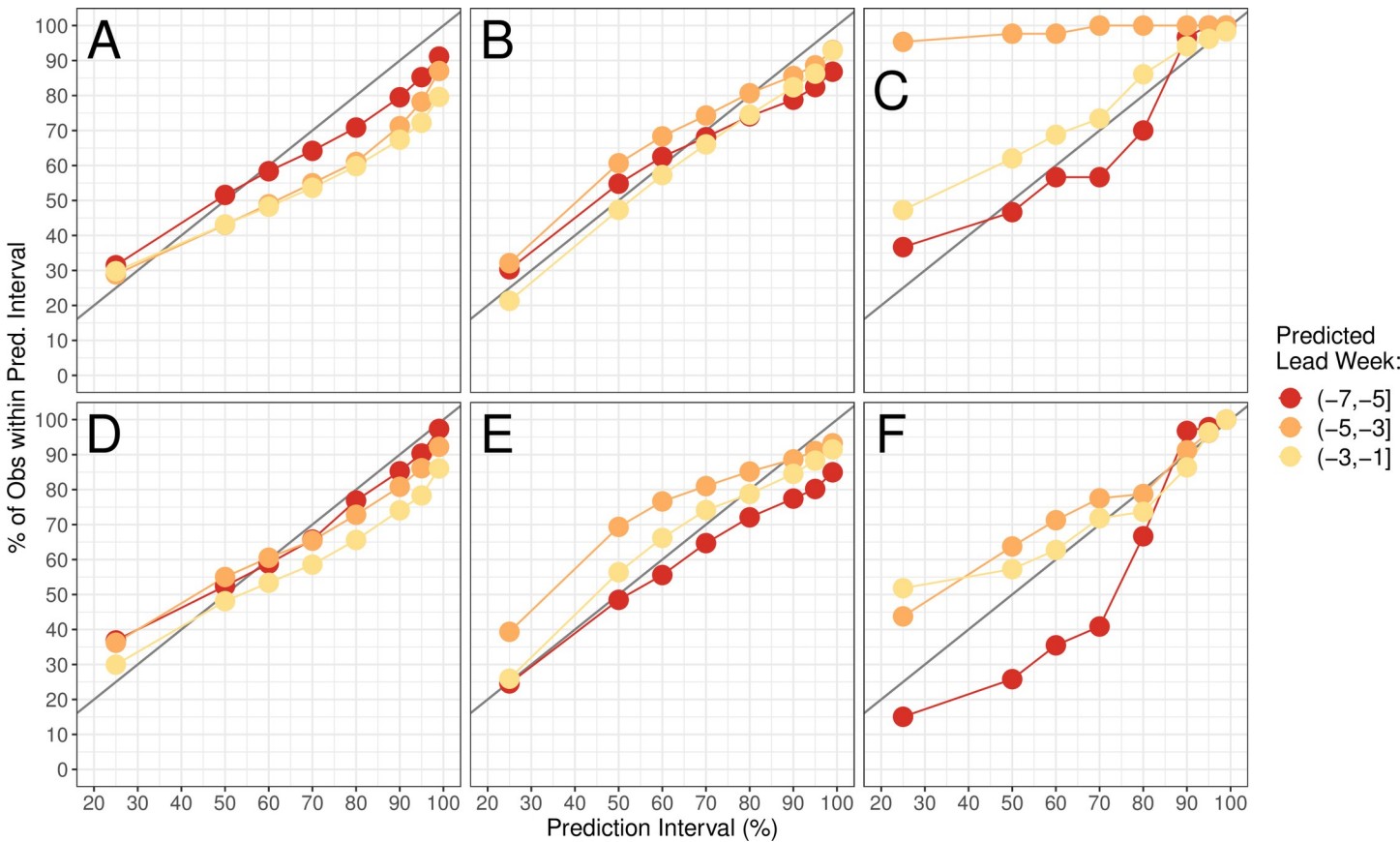

**Fig 4. Retrospective forecast calibration for the network (A-C) and isolated (D-F) models.** Points show the proportion of observed values for peak timing (A and D), peak intensity (B and E), and onset timing (C and F) that fell within the 25%, 50%, 80%, 90%, 95%, and 99% prediction intervals of 300 ensemble members. Colors represent the predicted lead to the peak (A-B, D-E) or to the onset (C and F). In a perfectly-calibrated model, we expect *n*% of observations to fall within the *n*% prediction interval; this situation is shown as a gray line for reference.

number of forecasts that predict any onset timing prior to the predicted onset was few, and the majority of countries and seasons had no or very few forecasts of onset timing until after onset occurred. When assessed statistically, we found no significant differences between the performance of the network and isolated models for peak timing, peak intensity, or onset timing.

Both the network and isolated models outperformed forecasts based on historical expectance alone. Low scores for the historical forecasts were primarily due to the fact that, because of heterogeneities in outbreaks season to season, observed values for a given season were rarely observed in the small number of other seasons included in our data. Thus, especially when few seasons of historical data are available, mechanistic forecasts like those presented in this work remain a promising tool. However, given that the metapopulation and isolated models rarely produced consistent and reliable forecasts of onset timing prior to outbreak onset, methods based on historical expectance should continue to be assessed for forecasting onset timing in particular. The availability of several more seasons of historical data would be of value here.

Model fit quality was significantly higher for A(H1) and tended to be lower for A(H3), but this did not seem to influence the (sub)type-specific forecasting results. While the network model appeared to improve forecasts of onset timing for A(H1) (S6 Fig), very few forecasts were generated (S3 Table), and the results were not statistically significant. The reason for this discrepancy in fit quality is unclear: all influenza (sub)types generally led to outbreaks in a

similar number of countries and had similar synchrony with regard to onset and peak timings by country. We also found no evidence of higher signal smoothness for outbreaks of A(H1). However, we did find that scaled outbreaks of A(H3) were significantly larger than outbreaks of the other two (sub)types. It is possible that the network model has more trouble fitting larger peaks, especially when it must simultaneously fit countries where no outbreak onset has occurred. (See S1 Text for descriptive statistics and data quality analyses by (sub)type.)

Overall, we found that despite the inclusion of human travel, the network model did not offer any substantial advantage when it came to influenza forecast generation. These results contrast previous findings, which showed that explicitly modeling travel (in particular commuting) between US states [27] and between New York City boroughs [26] significantly increased forecast accuracy, especially accuracy for onset timing predictions. While it is not possible to determine the exact roots of this discrepancy, it is likely that differences in data quality play a large role. In particular, we point to data quality issues that hinder comparison between countries. While the influenza data used in this study were likely noisier and more prone to missingness than the US data, we have shown that this is not necessarily associated with lower forecast accuracy in the isolated model [29] or in the network model (see S1 Text). That said, we note here that noisiness in the data from Luxembourg was associated with lower-quality model fit (see "Model Fit" above and S1 Text). It is possible that, while the impact of noisy data may be small on the level of the individual country, the influence of several countries is amplified when all countries must be fit simultaneously.

Likely more important, however, is the requirement that we have some idea of the relative intensity of influenza activity by country in order to properly model influenza transmission between countries. Unfortunately, the syndromic data as reported to FluID rarely have reliable denominator data; countries generally do not report the number of total visitors to sentinel facilities, and even reports of the total number of people within surveillance catchment areas are either missing or inconsistent over time. For this reason, we are unable to calculate rates of ILI or ARI.

Furthermore, although surveillance levels do vary by state within the US, surveillance systems are likely much more variable by country in Europe [43]. Even if it were possible to calculate rates of syndromic+ cases, heterogeneities and biases in surveillance strategies mean that these country-specific rates may not be comparable to rates from other countries. For example, direct comparison is not possible between countries collecting ILI versus ARI data, or collecting virologic data from all patients versus only those experiencing severe symptoms. These differences, compounded with the lack of denominator data, likely make it difficult to properly simulate and forecast international transmission in Europe. While we attempted to remedy this issue using scaling factors, this approach imperfectly accounts for multiple, possibly shifting, heterogeneities between locations (see S1 Text), and may be insufficient for the model-assimilation system developed here.

The extent to which these data quality issues reflect how the data are collected at the country level, or how the data are reported to the WHO, is unclear. Although a well-established network of laboratories reporting virologic data to the WHO exists [44], the push for centralized collection of syndromic data is recent [45,46], and reporting of data to FluNet and FluID remains voluntary. It is likely that many countries do not report all of the data they collect. However, obtaining and formatting data from several countries independently is time-consuming and, especially in the case of an emerging pandemic, impractical. Thus, barriers to timely and skillful forecasts of influenza transmission throughout Europe exist due to how data are collected, as well as how and whether data are reported to central databases.

In order to improve model performance, we recommend that, at minimum, denominator data for sentinel surveillance efforts be reported, preferably as the number of total visits (not

influenza-specific visits) made to sentinel sites. We also recommend that, where possible, countries use ILI rather than ARI, as it is more specific, and we have previously shown that ILI data yield more skillful forecasts of influenza activity [29]. While it would of course be helpful if surveillance systems were more similar country to country, we recognize that the ideal surveillance strategy for a country will depend on factors such as country size, healthcare system, and goals related to surveillance. The WHO does offer extensive guidance on improving influenza surveillance systems, particularly syndromic surveillance systems, which historically have been less developed than virologic surveillance systems, and are likely responsible for the majority of data quality issues encountered in this work [46]. We note there was a significant push to improve syndromic surveillance and reporting in the wake of the 2009 influenza pandemic [45,47]. Thus, we expect that improvement is indeed possible, but may depend on the extent to which influenza is seen as a public health priority.

We also note that the commuting data used here are likely of lower quality than those used in the United States. In the US, commuting data are captured by government census. While the European Labour Force Survey uses standardized definitions and questionnaires in order to bolster comparability between countries, data collection is still the responsibility of individual countries, meaning that data collection is inherently less centralized and standardized than in the US [48]. Furthermore, as we note in the Materials and Methods above, data below certain thresholds are not reported to users at all. However, because commuting along these routes is minimal, we do not expect this missingness to have greatly impacted results. Overall, we expect that the network model's inability to improve forecast accuracy is primarily driven by low-quality influenza, and not travel, data.

It is also possible that the model itself is not properly specified for exploring influenza transmission in Europe. In other words, while a model incorporating commuting data between locations may be appropriate for interstate influenza transmission in the US, commuting and air travel may not be as important at driving cross-border transmission in Europe. Despite high rates of cross-border commuting at border regions, international commuting in Europe still only accounts for 0.9% of commuting overall [49]. Influenza transmission may therefore rely more on non-commuting train and automobile traffic, neither of which are captured in our model. More robust data on train travel, and especially on automobile travel across country borders, could contribute to a better specified model, able to anticipate outbreak onsets based on activity in countries heavily linked by train routes and roads. Alternatively, commuting may be an important driver of influenza transmission in border regions, but country-level influenza overall may be much more heavily driven by travel patterns within the country itself. Models focusing on travel on this smaller sub-national scale may therefore have more success forecasting country-level influenza transmission.

Finally, we note that, as with any model, we are unable to capture all features of the system we seek to model. $R_0$ is a composite parameter that takes into account the intrinsic transmissibility of a given pathogen, as well extrinsic factors driving transmission, such as environmental factors or contact networks [50,51]. Our network model only allows differences in $R_0$ by country on the basis of country-level absolute humidity. However, we recognize that contact patterns may be driven by population density, age structure, school holiday schedules, and various other demographic and cultural factors not captured here that nonetheless can vary greatly by country [52–54]. While it may be possible to implicitly consider these differences in our model by allowing $R_{0max}$ and $R_{0diff}$ to vary by country, explicit consideration of these differences is problematic because we do not understand exactly how they may influence influenza transmission. While it can be tempting to add complexity to a model in an effort to increase its fit of observations, adding details without understanding their real-life impact has the potential to backfire, leading to overfit results that are relevant only to the model

population and not to reality [55,56], and which may also corrupt forecast accuracy. Future work should attempt to better understand the drivers of international influenza spread not just between European countries, but on a variety of spatial scales, both smaller and larger.

## Conclusions

Here we present a novel network model of influenza transmission among twelve European countries, and test the model's ability to improve influenza forecasting accuracy over isolated, country-level models. While the network model system was unable to improve forecasts in most circumstances, the success of similar network models in the United States suggests that this could be a powerful tool for improving influenza forecasts if data quality issues were to be addressed. We identify key opportunities for improvement in data collection and sharing that may allow for success in the future. In the meantime, future work should focus on better understanding the various drivers of international influenza transmission in Europe and globally, so that models can better account for these relevant factors.

## Supporting information

**S1 Text. Supplementary methods and results.**
(PDF)

**S1 Dataset. Processed syndromic+ observations from 12 countries (H1N1).** Observations are scaled according to values in S1 Table. Note that scaled values are only available for seasons during which H1N1 influenza was forecasted; values in other seasons were set to be NA.
(CSV)

**S2 Dataset. Processed syndromic+ observations from 12 countries (H3N2).** Observations are scaled according to values in S1 Table. Note that scaled values are only available for seasons during which H3N2 influenza was forecasted; values in other seasons were set to be NA.
(CSV)

**S3 Dataset. Processed syndromic+ observations from 12 countries (B).** Observations are scaled according to values in S1 Table. Note that scaled values are only available for seasons during which B influenza was forecasted; values in other seasons were set to be NA.
(CSV)

**S4 Dataset. Processed absolute humidity climatologies for 12 European countries.** Columns contain the average absolute humidity in a country, weighted by population size, on days 1–365 of the year.
(CSV)

**S1 Fig. Syndromic+ data by country and subtype over the course of the study period.** The beginning of each season (week 40) is marked on the x-axis. Because France shifted from reporting ARI to ILI prior to the 2014–15 season, ARI data from the 2012–13 and 2013–14 seasons are shown as an inset.
(SVG)

**S2 Fig. Ratio of observational error variance to prior ensemble variance over time.** Ratios are plotted separately for the (A) metapopulation and (B) isolated models. The black line shows the median ratio over all countries, (sub)types, seasons, and runs; 50% and 95% credible intervals are shaded in dark and light gray, respectively. The y-axis is plotted on a log scale.
(SVG)

**S3 Fig. Cross-ensemble Pearson's correlation coefficients between observed state variables and unobserved model states and parameters in the network model.** Correlations with modeled incidence at the country level are displayed for (A) $S$ among those who live and work in the corresponding country, (B) $I$ as in (A), (C) $newI$ as in (A), (D) $L$, (E) $D$, (F) $R_{0max}$, (G) $R_{0diff}$, and (I) $air_{Adj}$. Black lines represent median correlation coefficients; 50% and 95% credible intervals are shaded in dark and light gray, respectively. Correlations are shown over the course of the influenza season (week 40 through week 20 of the following year), averaged over all countries, (sub)types, seasons, and model runs.
(SVG)

**S4 Fig.** Model fitting to observed A(H3) (A) and B (B) influenza observations throughout the 2012–13 season. Scaled syndromic+ observations are shown as points, while blue lines represent inferred model incidence. Each line represents the mean inferred model incidence over 300 ensemble members for one of five runs, each with different initial conditions. The shaded gray areas represent the 95% confidence intervals for each of the runs, calculated using the ensemble means and standard deviations. (A) RMSE ranged from 45.78 (ES) to 353.73 (CZ), with mean = 171.64 (median = 168.16). Pearson's correlation coefficients ranged from 0.300 (PL) to 0.977 (FR), with mean = 0.867 (median = 0.903). (B) RMSE ranged from 105.60 (PL) to 1145.5 (LU), with mean = 384.00 (median = 318.20). Pearson's correlation coefficients ranged from 0.794 (LU) to 0.984 (AT), with mean = 0.943 (median = 0.972).
(SVG)

**S5 Fig. Retrospective forecast accuracy by observed lead week.** Results are shown both before (A-C) and after (D-F) removing season-country-subtype-week pairs for which either the network or isolated model predicted no onset, as well as when all forecasts, including those predicting no onset, are maintained (G-I). Log scores are shown for peak timing (A, D, and G), peak intensity (B, E, and H), and onset timing (C, F, and I). Mean log scores for the network model are shown in red, and scores for the isolated models are shown in blue. The size of the points represents the number of forecasts generated at a given lead week.
(SVG)

**S6 Fig.** Log scores by predicted lead week for forecasts of peak timing (A-C), peak intensity (D-F), and onset timing (G-I), separated by (sub)type. Network results are shown in red, and results from the isolated model are shown in blue. Point size represents the number of forecasts generated at a given lead week for which an onset was predicted.
(SVG)

**S7 Fig. Log scores by predicted lead week for forecasts of peak timing and peak intensity, shown separately for each of the twelve countries in the network model.** Network results are shown in red, and isolated in blue. Point size represents the number of forecasts generated that predicted an onset.
(SVG)

**S8 Fig. Retrospective forecast accuracy by mean absolute (percentage) errors (MA(P)E).** Results are shown by predicted lead week for peak timing (A), peak intensity (B; MAPE), and onset timing (C). Network results are in red and isolated in blue. Point size represents the number of forecasts generated for which onsets were predicted.
(SVG)

**S9 Fig. Retrospective forecast accuracy by MAE/MAPE by predicted lead week, separated by influenza (sub)type.** Results are shown for peak timing (A-C), peak intensity (D-F), and onset timing (G-I). Network results are in red and isolated in blue. Point size represents the

number of forecasts generated for which an onset was predicted.
(SVG)

**S10 Fig. Histograms of forecast error for peak timing (A) and intensity (B), shown by (binned) predicted lead week.** Peak timing error is shown with bins of size 1 week, and relative peak intensity error is shown with bins of size 0.1. The y-axis represents the proportion of forecasts falling into a given bin. Results from the network model are shown in red; results from the isolated model are shown in blue.
(SVG)

**S11 Fig. Fittings of model parameters $D$ (A), $R_{0max}$ (B), and $R_{0diff}$ (C) over time.** Each panel in (A), (B), and (C) represents one of each of five synthetic outbreaks, as described in S4 Table. Fits at each week are shown as gray points; solid black lines represent the true values.
(SVG)

**S12 Fig. Histograms of the relative error of fits of $\beta$ (A), $R_0$ (B), and $R_{eff}$ (C) at four timepoints over all countries and synthetic outbreaks.** The y-axis shows the proportion of fits falling into each bin of size 0.05. Dotted lines show relative error equal to 0, for reference.
(SVG)

**S13 Fig. Sensitivity (A) and specificity (B) of binary onset predictions by calendar week.** Network model results are in red and isolated are in blue. Results are limited to weeks on or after the first observed outbreak onset of each season and (sub)type. Only one season-(sub)type began on week 47 (2016–17, A(H3)), and all countries had observed outbreaks, so specificity could not be calculated for this week.
(SVG)

**S14 Fig. Values of model states and parameters, as fit by the network model, shown separated by (sub)type.** Fittings are shown for $S0$ (A), maximum $Reff$ (B), $R_0$ (C), $D$ (D), $L$ (E), and $air_{Adj}$ (F). Boxes extend from the first to the third quartile, with a horizontal line marking the median, while values more than 1.5 times the interquartile range below the first quartile or above the third quartile are shown as points.
(SVG)

**S1 Table. Scaling factors by country and (sub)type.** Note that France switched from preferentially collecting ARI to ILI data prior to the 2014–15 season; scaling factors for France are listed as ARI/ILI.
(PDF)

**S2 Table. Number of country-season-(sub)type-week forecast pairs removed from consideration prior to conducting Friedman tests, by observed peak and onset lead week.** Pairs are removed when either the network or isolated model produces no forecasts with any predicted onset. Final numbers reflect the number of forecasts included in the Friedman test analyses from the main text, and in S3D–S3F Fig.
(PDF)

**S3 Table. Number of forecasts predicting any onset by predicted onset lead week, separated by (sub)type.**
(PDF)

**S4 Table. Parameters values used to generate synthetic outbreaks, and the number of countries with outbreak onsets.**
(PDF)

## Acknowledgments

The authors would like to thank Ralf Strobel for helping to improve model efficiency and runtime.

## Author Contributions

**Conceptualization:** Sarah C. Kramer, Jeffrey Shaman.

**Data curation:** Sarah C. Kramer.

**Formal analysis:** Sarah C. Kramer.

**Funding acquisition:** Sarah C. Kramer, Jeffrey Shaman.

**Supervision:** Sen Pei, Jeffrey Shaman.

**Visualization:** Sarah C. Kramer.

**Writing – original draft:** Sarah C. Kramer.

**Writing – review & editing:** Sen Pei, Jeffrey Shaman.

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
