## [Decision Letter · Decision Letter 0]

13 May 2020

Dear Ms. Kramer,

Thank you very much for submitting your manuscript "Forecasting influenza in Europe using a metapopulation model incorporating cross-border commuting and air travel" for consideration at PLOS Computational Biology.

As with all papers reviewed by the journal, your manuscript was reviewed by members of the editorial board and by several independent reviewers. In light of the reviews (below this email), we would like to invite the resubmission of a significantly-revised version that takes into account the reviewers' comments.

We cannot make any decision about publication until we have seen the revised manuscript and your response to the reviewers' comments. Your revised manuscript is also likely to be sent to reviewers for further evaluation.

Sincerely,

Benjamin Althouse

Associate Editor

PLOS Computational Biology

Virginia Pitzer

Deputy Editor

PLOS Computational Biology

Reviewer's Responses to Questions

**Comments to the Authors:**

Reviewer #1: This report compares ensemble predictions of influenza evolution for 12 European countries using a SIRS model that includes information about commuting and air travel and one that does not. The conclusions are that there is no apparent improvement from adding the travel information.

I have limited expertise in this type of model or application, but significant expertise in ensemble filtering and the statistical science of quantifying prediction skill and uncertainty. My comments will focus almost entirely on concerns about the clarity of the description of the filtering method and specifics of the forecast system design and validation.

My opinion is that the manuscript in its present form is not suitable for publication. There are four major issues that need addressed and it is unclear to me how difficult that will be. In particular, the filtering algorithm is not clearly described in the text and equations and fundamental filtering challenges are not mentioned or addressed. The algorithm also appears to withhold large sets of forecasts depending on a priori measures of forecast quality that are either not possible to assess in real time or depend on forecast outcome. It is nearly impossible to assess the quality of forecasts when this is done. Finally, the models appear to be tuned without cross validation using a very small set of forecast years. It is possible that all of these problems can be addressed by clarifying the text. However, I must recommend rejection for now without knowing that is the case.

Major comments:

1. The description of the EAKF is insufficiently clear and complete. Equation (3) is correct, however, X and O are not clearly defined. I assume that X includes the state variables and the parameters, but a list of the parameters that are estimated is never given and the exact size and definition of the state is not clearly state. In addition, the functional relation between O and X is not clearly presented; this would be referred to as a forward operator in most EAKF applications. The section ‘choice of scaling factors’ appears to be defining the forward operators, but it is insufficiently clear. At the end, there should be equations saying X=, and O=f(X) where f is clearly defined.

2. The EAKF suffers from two types of error, insufficient prior variance and overestimated correlations. These problems must be addressed by some form of localization and inflation respectively. I’m willing to give the authors some slack on localization given that they use fairly large ensembles (I think, although there is the question of whether they are rejected even in the EAKF analysis). Still, they should make some sort of statement about the magnitude of correlations between their observations and state/parameters to give confidence that the assimilation is meaningful. Inflation is a much bigger issue here since there is clearly large model error and parameters without evolution equations are being estimated. This will certainly lead to insufficient variance and almost certainly to filter collapse/divergence for the application. The authors make only one passing mention of this, provide no evidence that it is not happening (and suggest that it is), and provide no discussion of why they are able to avoid dealing with a problem that is almost universal. I think it is unlikely that this EAKF application is statistically valid without inflation and I need some sort of evidence of consistency between prior ensembles and observations to convince me otherwise. There is a vast literature on this subject and something needs to be cited.

3. The authors appear to withhold forecasts based upon criterion that either cannot be computed in real-time or that are conditional on the desired result. This generally results in residual forecast sets that cannot be evaluated with simple statistical tests. The first example occurs in the section ‘choice of seasons’. Forecasts are not made if the positivity rate is low. This can be computed in realtime, but why exclude these forecasts? If the model has predictive capability, it should respond by keeping rates low, unless they are going to get high. What can we conclude about the model if it is not able to do this correctly? A more serious example occurs around line 335. Forecasts are only compared for initial conditions in a fixed interval around a specific verification criterion. It appears that this condition cannot be computed in real-time in some instances. Whether that is the case or not, it seems false positives are excluded from independent evaluation as well as true negatives. Again, it is unclear why one would exclude these from a valid prediction model and it becomes extremely difficult to evaluate the results. The conditional probability is very difficult to even define, much less fairly score. The authors should give a very clear explanation of why this is a meaningful prediction test and how they account for this conditioning in the scoring. They should also provide clear quantitative information about how many forecasts are discarded for each of these criterion (and any others). If the fraction is large, as it seems to be from qualitative discussion, I do not think it is possible to make meaningful statements about the models’ absolute or relative predictive capabilities. Please convince me otherwise.

4. Tuning models without cross validation is generally unacceptable, especially for a sample with so few independent events (years). Again, the text is insufficiently clear, but it appears that the model is tuned with dependent data in a variety of ways. The clearest example seems to be in the section on choosing the scaling factors. I believe that there are also problems with validating forecast for time of peak intensity by lead weak which can only be determined once the peak week is known. This should be carefully discussed.

5. This isn’t really a major issue, and it may have to do with this journal. However, it is incredibly annoying to not have figure captions collocated with the figures. If at all possible, duplicate the figure captions and place them near the figures, or if the journal allows, imbed the figures in the text with the captions.

Minor stuff:

1. Line 149: What does it mean to “maintain only countries…” Do you mean you only modeled for these countries?

2. Line 156, 57: Specify what distribution you drew from. Uniform? EAKFs would be much happier with a normal distribution when possible.

3. Line 175: These are not technically ‘observations’. They are a gridded analysis that gives a lot of weight to climatology.

4. Line 208: I do not believe you have defined AH yet.

5. Line 239: It would be clear to also mention how you initialize the state variables at this point.

6. Line 240: Again, define the distribution. I assume it is uniform between these bounds. Note again that the EAKF is really designed to work with normal distributions. It is always a challenge to deal with parameter estimation, especially if the quantity is bounded, and you should probably note this or provide a reference.

7. Line 242, also state the variables (like S) from the previous equation here rather than just text names.

8. Line 243: Define the system parameter set very clearly here and include the variables from the previous equations.

9. Line 251: This is an inaccurate description of what is in X (see major point). State precisely what the vector X includes. A set of state variables for each population subdomain and also parameters? Be mathematically precise.

10. Line 265: Weeks 44 through 69 did not make sense to me. Clarify.

11. Line 267: You should specify why you chose 300 members. This might fit in with a sentence addressing the absence of localization in your EAKF implementation.

12. Line 271: This is related to a major concern. One would expect you to forecast that it has little impact rather than concluding it has little impact and not forecasting it.

13. Line 280-81. This statement is fundamentally incorrect as your equation (3) points out. Here, I assume that the “data” means observations and that the observations are, of course, not of your specific variables in X. You need to define O=f(X) in this section (see major comment).

14. Line 282: You are not being consistent with the use of the terms ‘observations’ and ‘data’. You have previously defined observations and have used O in equation (3). Here, I believe ‘data’ would be better replaced by ‘observation’. Whatever the case, be clear and consistent.

15. Line 285: This appears to be where cross validation would be appropriate. Otherwise, you will be overfitting by including the year being forecast in the derivation of the calibrating statistics.

16. Line 303-04: I have no idea what this sentence is about. What is ‘training’ data. What were these weeks? How does this relate to the EAKF analyses?

17. Line 325-29. This is unclear. Obviously, you can generate forecasts for anything that is a function of X at any time after the initial time. You can only validate forecasts when you have observations. An isolated model can also predict anything at any lead time. Are you making a statement that you don’t generate forecasts from times with no observations at the intial/analysis time? Please clarify.

18. Line 340-41. This is a very odd statement. You are conditioning your interest on the forecast output. This is statistically unusual; could you even write down a conditional probability expression for this? I also think it’s inherently false. If you are interested in forecasts of onset, then no onset is the negation of that so it provides information.

19. Line 344-347. This is unclear. Sounds like you didn’t compare the same forecasts for each country for your multi-country system. That seems unusual.

20. Line 377: ‘closely fitting’ This statement is meaningless. Make a quantitative statement or leave it out. Same for line 379 “appear to be well-fit”

21. Line 385: Only mention of filter divergence (see major comment). You can evaluate quantitatively if the filter is diverging, not only at the end, but also earlier. Clearly needs to be done to have any trust in the system.

22. Line 395-96. This was not clear enough. I am not sure if these are forecasts, analyses, free runs.

23. Line 408-409. This was perhaps the most troubling sentence about the exclusion of some forecasts. Those that do not predict an onset are ‘unreliable’. If you mean this, that the model cannot predict cases where there is no onset, it is very difficult to envision a dynamical system where a model can do something meaningful about predicting an event, but not its inverse. I’d need to see a discussion of why this could be the case to have any confidence in the results.

24. Line 412: This needs to be much more quantitative. A handful of dots covering large ranges is insufficiently informative when you are talking about rejecting many forecasts. I suggest a plot that includes two vertical axes, one plotting the metric and the other plotting the number of retained forecasts.

25. Line 447. Another troubling sentence. If you are really only retaining a few forecasts, and rejecting most, it is difficult to understand how the predictions can be valid.

26. Line 454-56. Comparing forecast error and spread would be a more traditional and robust statistic than a range. Motivate why you chose the range.

27. Line 462: Yet another concern about rejecting forecasts.

28. Line 469: Units (weeks) should be made clear in the figure.

29. Line 472: Grey line was very hard to see.

Reviewer #2: Kramer, Pei, and Shaman extend the SIRS-EAKF framework to predict the onset timing, peak timing, and peak intensity of several subtypes of influenza in 12 European countries, and evaluate whether incorporating travel patterns between the countries improves forecasts. While their forecasts incorporating travel do not substantially improve forecasts, as seen in the group’s prior efforts for the US and New York City, the authors provide a remarkably thorough discussion of the model assumptions and the shortcomings of the data.

All of the major claims comparing the network model (using travel data) and the isolated models are either well-founded or addressed in the discussion. One quibble I might have would be that no baseline model is used as a general comparison for forecast accuracy (e.g. an ARIMA or a model of historical averages). However, the authors avoid making any statements about the overall accuracy of their models and correctly restrict their statements to comparisons between the presented models.

The authors provide the data (either explicitly in the supplement or by providing the source in the text) and methods required to run the analysis, though I would also appreciate it if they made the corresponding code freely available as well.

**Have all data underlying the figures and results presented in the manuscript been provided?**

Reviewer #1: Yes

Reviewer #2: None

PLOS authors have the option to publish the peer review history of their article (what does this mean?). If published, this will include your full peer review and any attached files.

Reviewer #1: No

Reviewer #2: Yes: Stephen A Lauer
---

## [Decision Letter · Decision Letter 1]

10 Aug 2020

Dear Ms. Kramer,

We are pleased to inform you that your manuscript 'Forecasting influenza in Europe using a metapopulation model incorporating cross-border commuting and air travel' has been provisionally accepted for publication in PLOS Computational Biology.

Best regards,

Benjamin Althouse

Associate Editor

PLOS Computational Biology

Virginia Pitzer

Deputy Editor

PLOS Computational Biology

Reviewer's Responses to Questions

**Comments to the Authors:**

Reviewer #1: I commend the authors for an exceptional response to my long-list of comments and questions. I always try to stay positive in my reviews, but noticed that I did get a bit snarky in a few comments here, so apologies. The revised manuscript is unusually high-quality and I have recommended acceptance. I still have serious concerns about the dangers of using forecasts that are selectively rejected, and suggest the authors think about this issue a bit more. However, the text is very clear about what is going on and the conclusions are about relative forecast capability here.

Reviewer #2: All concerns were addressed. Please make sure that the link to the Github repository with the analysis code and data is mentioned within the methods section of the manuscript.

**Have all data underlying the figures and results presented in the manuscript been provided?**

Reviewer #1: Yes

Reviewer #2: Yes

PLOS authors have the option to publish the peer review history of their article (what does this mean?). If published, this will include your full peer review and any attached files.

Reviewer #1: No

Reviewer #2: **Yes: **Stephen A Lauer

---

## [Editor Report · Acceptance letter]

6 Oct 2020

PCOMPBIOL-D-20-00498R1 

Forecasting influenza in Europe using a metapopulation model incorporating cross-border commuting and air travel

Dear Dr Kramer,

I am pleased to inform you that your manuscript has been formally accepted for publication in PLOS Computational Biology. Your manuscript is now with our production department and you will be notified of the publication date in due course.

With kind regards,

Sarah Hammond
